# A proteomic network approach across the ALS-FTD disease spectrum resolves clinical phenotypes and genetic vulnerability in human brain

Mfon E Umoh[1,2,†], Eric B Dammer[2,3,†], Jingting Dai[2,3], Duc M Duong[2,3], James J Lah[1,2], Allan I Levey[1,2], Marla Gearing[1,2,4], Jonathan D Glass[1,2,4,*] & Nicholas T Seyfried[1,2,3,**]

## Abstract

Amyotrophic lateral sclerosis (ALS) and frontotemporal dementia (FTD) are neurodegenerative diseases with overlap in clinical presentation, neuropathology, and genetic underpinnings. The molecular basis for the overlap of these disorders is not well established. We performed a comparative unbiased mass spectrometry-based proteomic analysis of frontal cortical tissues from postmortem cases clinically defined as ALS, FTD, ALS and FTD (ALS/FTD), and controls. We also included a subset of patients with the C9orf72 expansion mutation, the most common genetic cause of both ALS and FTD. Our systems-level analysis of the brain proteome integrated both differential expression and co-expression approaches to assess the relationship of these differences to clinical and pathological phenotypes. Weighted co-expression network analysis revealed 15 modules of co-expressed proteins, eight of which were significantly different across the ALS-FTD disease spectrum. These included modules associated with RNA binding proteins, synaptic transmission, and inflammation with cell-type specificity that showed correlation with TDP-43 pathology and cognitive dysfunction. Modules were also examined for their overlap with TDP-43 protein–protein interactions, revealing one module enriched with RNA-binding proteins and other causal ALS genes that increased in FTD/ALS and FTD cases. A module enriched with astrocyte and microglia proteins was significantly increased in ALS cases carrying the C9orf72 mutation compared to sporadic ALS cases, suggesting that the genetic expansion is associated with inflammation in the brain even without clinical evidence of dementia. Together, these findings highlight the utility of integrative systems-level proteomic approaches to resolve clinical phenotypes and genetic mechanisms underlying the ALS-FTD disease spectrum in human brain.

**Keywords** C9orf72 expansion mutation; mass spectrometry; neurodegeneration; protein co-expression; TDP-43

**Subject Categories** Neuroscience; Post-translational Modifications, Proteolysis & Proteomics

## Introduction

Amyotrophic lateral sclerosis (ALS) and frontotemporal dementia (FTD) are clinically distinct neurodegenerative diseases that are connected by genetic and pathological overlap (Fecto & Siddique, 2011; Ferrari et al, 2011; Achi & Rudnicki, 2012). ALS patients present with muscle weakness and spasticity associated with degeneration of motor neurons in the motor cortex, brainstem, and spinal cord that ultimately leads to death. In contrast, patients with FTD display cognitive dysfunction associated with degeneration of neurons in the frontal and temporal lobes of the brain. Despite being clinically distinct, 15% of individuals presenting with FTD also have ALS, whereas 30% of individuals with ALS will develop FTD (Lomen-Hoerth, 2011). This implies that these two neurodegenerative diseases are part of a shared clinical spectrum.

In addition to their clinical overlap, most cases of ALS and FTD display pathological accumulation of TAR-DNA binding protein (TDP-43), a ubiquitously expressed nuclear DNA/RNA binding protein that is cleaved, phosphorylated, and aggregated in the cytoplasm in disease (Neumann et al, 2006). Ninety-seven percent of ALS cases display phosphorylated TDP-43 pathology in the brain and/or spinal cord, while 50% of FTD cases display this pathology throughout the brain (Radford et al, 2015), defining these diseases as TDP proteinopathies. Many individuals with ALS and FTD also share a positive family history of disease (Fong et al, 2012). The largest proportion of inherited cases (40% ALS and 25% FTD) are caused by hexanucleotide $G_4C_2$ repeat

1   Department of Neurology, Emory University School of Medicine, Atlanta, GA, USA
2   Center for Neurodegenerative Diseases, Emory University School of Medicine, Atlanta, GA, USA
3   Department of Biochemistry, Emory University School of Medicine, Atlanta, GA, USA
4   Department of Pathology and Laboratory Medicine, Emory University School of Medicine, Atlanta, GA, USA
    *Corresponding author. Tel: +1 404 727 3275; E-mail: jglas03@emory.edu
    **Corresponding author. Tel: +1 404 712 9783; E-mail: nseyfri@emory.edu
    †These authors contributed equally to this work

expansions in the C9orf72 gene, which notably were identified from families with co-occurring ALS and FTD (ALS/FTD) (DeJesus-Hernandez *et al*, 2011; Renton *et al*, 2011; Majounie *et al*, 2012; Radford *et al*, 2015). Although the function of the C9orf72 protein is not yet known, there are several theories regarding how the C9orf72 mutation leads to ALS and FTD (Zhang *et al*, 2012; Farg *et al*, 2014). For example, loss of C9orf72 protein expression is thought to inhibit autophagy and promote neuroinflammation (Farg *et al*, 2014; Webster *et al*, 2016), whereas expression of C9orf72 gene products could cause toxicity via nuclear sense and antisense repeat-containing RNAs that sequester RNA binding proteins (abnormal RNA metabolism) or by non-canonical repeat-associated non-ATG (RAN) translation of dipeptide repeat proteins that aggregate and block nuclear pores (Gitler & Tsuiji, 2016). Notably, this genetic mutation bridges these two diseases that clinical and pathological overlaps have previously connected. However, the underlying molecular basis of the ALS-FTD clinical and pathological spectrum is not well established. It is also unclear why patients with the same C9orf72 genetic expansion get either or both of these disparate diseases. Using our knowledge of the shared clinical, pathological, and genetic features characterizing ALS and FTD, a systems-level proteomic analysis of both sporadic and genetic (C9orf72) cases comprising the ALS-FTD spectrum was conducted to determine common and distinct pathways that contribute to the onset and development of dementia.

Co-expression network analysis has been used to define modules of co-expressed genes or proteins linked to specific cell types, organelles, and biological pathways (Miller *et al*, 2008; Oldham, 2014; Seyfried *et al*, 2017). Assessing co-expression of proteins within samples and relating co-expression modules to clinical and pathological endophenotypes can be defined utilizing weighted co-expression network analysis (WGCNA), where the most centrally connected proteins in a module often act as key drivers (Zhang & Horvath, 2005; Oldham *et al*, 2008). We recently reported the first large-scale proteomic and transcriptomic multinetwork analysis in human postmortem brain from both asymptomatic and symptomatic Alzheimer's disease (AD) (Seyfried *et al*, 2017). This work revealed that several protein-driven processes related to cognitive decline are distinct from networks in human AD transcriptome. Moreover, analysis of the proteome is particularly relevant since neurodegenerative diseases are collectively viewed as proteinopathies defined by their association with the aggregation and accumulation of misfolded proteins (Golde *et al*, 2013).

Here, we report the first unbiased proteomic analysis of postmortem cortical tissues from clinically characterized ALS, FTD, ALS/FTD, and healthy disease controls. A subset of C9orf72 expansion-positive (C9Pos) cases was also included. WGCNA revealed 15 modules of co-expressed proteins, eight of which were significantly different across the ALS-FTD disease spectrum. These included modules associated with RNA binding proteins, synaptic transmission, inflammation, and cell-type specificity (neuronal, microglial, and astrocytic) that showed strong correlation with TDP-43 pathology and cognitive dysfunction. Compared to sporadic ALS patients, C9Pos ALS cases showed increased levels for a protein module associated with astrocytic and microglial markers, which supports a hypothesis that links the C9orf72 mutation to neuroinflammation.

# Results

## Proteomic signature of human brain classifies cases by clinical phenotypes

This study offers an in-depth analysis of protein changes in the frontal cortex of patients across the ALS-FTD disease spectrum, which resulted in the final quantification of 2,612 protein groups mapping to 2,536 unique gene symbols across all samples (Datasets EV1 and EV2). Our goal was to compare ALS patients with and without clinical dementia, so our experimental case selection grouped samples based on clinical phenotype. As FTD is clinically heterogeneous, we limited that group to include only individuals with the pathological diagnosis of frontotemporal lobar degeneration with TDP-43 inclusions (FTLD-TDP), which accounts for the majority of overlap with ALS and FTD (Neumann *et al*, 2009). The frontal cortex was chosen because it was a likely site for discriminating proteomic differences in ALS patients with and without dementia, and represents an area in which TDP-43 pathological burden has been mapped in FTD patients (Brettschneider *et al*, 2014). In our analytic pipeline, protein expression values are adjusted for age, sex, and postmortem interval (PMI) covariance prior to downstream differential or co-expression analyses (Fig EV1). Differential expression by ANOVA comparisons of each of the groups yielded subsets of significantly altered proteins across controls and disease groups (Dataset EV2). The number of significantly altered proteins increased sequentially when comparing controls to ALS, ALS-FTD, and FTD, respectively, which is likely due to the presence of clinical dementia and an increase in associated pathological burden in the frontal cortex (Fig 1A). For supervised clustering analysis, we selected 165 proteins that had at least two ANOVA Tukey pairwise comparisons of high significance ($P < 0.01$) among the six possible comparisons across the clinical groups. The over-represented gene ontology (GO) terms within these 165 information rich proteins included cytoplasmic vesicle part, clathrin coat, and plasma membrane (Dataset EV3). Using these differentially expressed protein signatures of disease, multidimensional scaling (MDS) was used to stratify the relatedness of individual cases, which revealed that clinical phenotypes indeed associate with proteomic signatures, as samples segregated into clusters representing clinical groups (Fig 1B). Also, ALS/FTD cases distributed between the ALS and the FTD clusters, supporting an underlying molecular and biological spectrum designated by protein expression that defines the clinical spectrum. Thus, this analysis highlights that protein signatures of ALS-FTD clinical phenotypes are related to differences in molecular pathways in the frontal cortex.

## Protein co-expression network analysis

Protein co-expression networks reflect relationships between protein pathways, cell types, and physically interacting proteins within modules. Network analyses revealed 15 modules of strongly co-expressed groups of proteins (Fig 2 and Dataset EV2). Each module is defined by an eigenprotein, the most representative weighted protein expression pattern across samples for a group of co-expressed proteins (Seyfried *et al*, 2017). Eight of these 15 modules, identified by numbers that correspond to a color, ordered 1–15, with

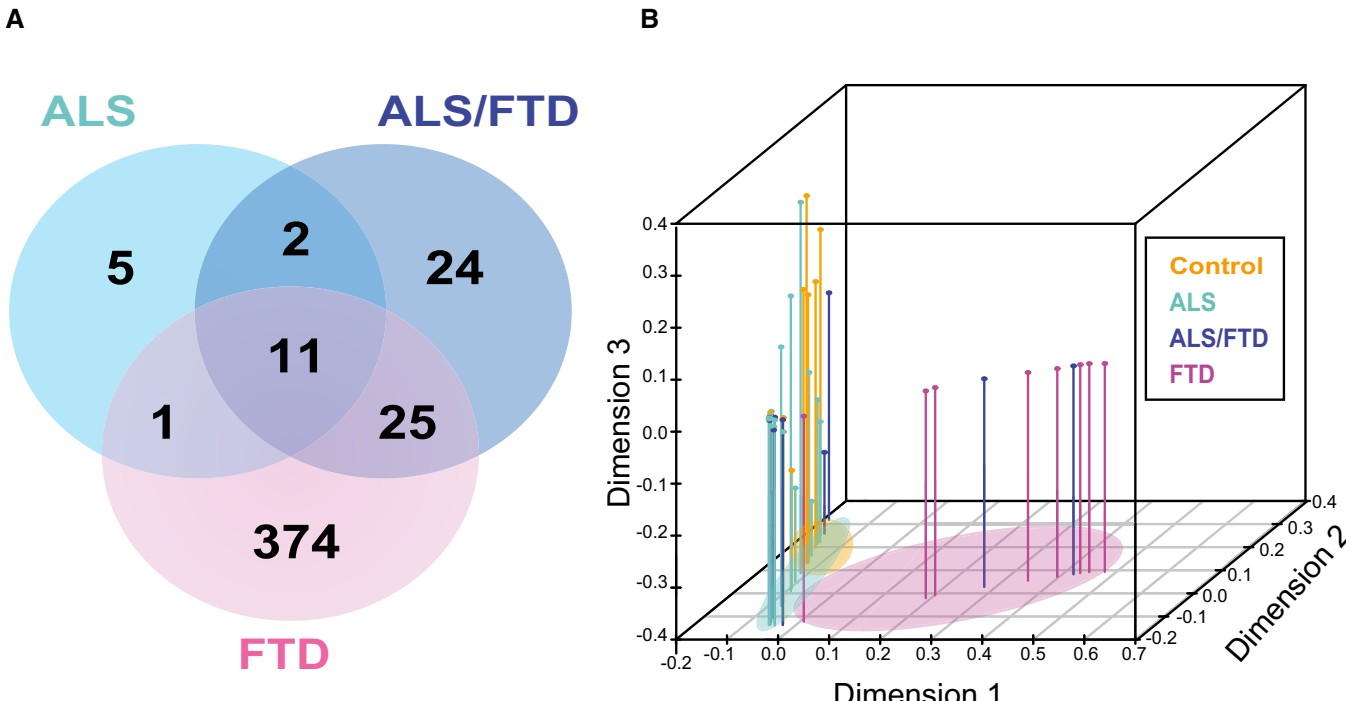

**Figure 1.  Segregation of ALS and FTD clinical phenotypes by proteomic signatures.**

A   Venn diagram showing a total of 442 unique proteins determined to be significantly altered by ANOVA followed by Tukey's *post hoc* test ($P \leq 0.01$) among three pairwise comparisons (i) ALS, (ii) ALS/FTD, and (iii) FTD versus control cases.

B   Supervised hierarchical clustering of 165 significant proteins altered in the frontal cortex across clinical phenotypes (Control, ALS, ALS-FTD, and FTD) was used as input for multidimensional scaling (MDS) analysis. Each dimension on the plot explains a larger proportion of variance of the dataset. Clinical groups are indicated by colors (orange—control, light blue—ALS, dark blue—FTD/ALS, and pink—FTD). Segregation of cases based on clinical grouping is indicated by colored ovals.

M1 (largest, 548 proteins) to M15 (smallest, 21 proteins), showed different patterns of co-expression across the four clinical groups. As expected, functional annotation of frontal cortex modules classified the proteome with specific gene ontologies and brain cell types among other biological sources of co-expression (Gaiteri *et al*, 2014). Of the significant modules, four (M2, M6, M9, and M10) showed increased expression in FTD cases compared to ALS and control cases (Fig 2A). These modules were enriched for RNA splicing (M2), response to biotic stimuli (M6), zinc ion binding (M9), and homeostatic processes (M10) (Dataset EV3). One of the significant modules, M15, enriched for blood microparticles and circulating immunoglobulin complexes, showed increased protein expression in all disease groups compared to controls, consistent with a common mechanism of blood–brain barrier breakdown in neurodegenerative diseases (Carvey *et al*, 2009; Seyfried *et al*,

2017) (Fig 2A). The remaining three significant modules (M1, M3, and M8) showed a decrease in protein expression in FTD cases compared to control and ALS cases; M1 and M8 were enriched in synaptic and neuronal proteins, while M3 was enriched with mitochondrial proteins (Fig 2A and Dataset EV3). Moreover, several modules showed significant correlation with clinical and pathological sample traits. Pathological sample traits characterize TDP-43 pathology, which is represented by the abundance of phosphorylated TDP-43 cytoplasmic inclusions in the frontal cortex (defined by the pTDP score) or label-free protein quantification (LFQ) of TDP-43 by mass spectrometry (Fig EV2); for example, M5, a module enriched with extracellular matrix and astrocyte proteins correlated closely to clinical and pathological traits (Fig 3A), M14, a module enriched with microtubule proteins and displaying dynactin as a hub protein, and optineurin, genes previously implicated in ALS

**Figure 2.  Integrated protein co-expression and differential expression across the ALS-FTD disease continuum.**

A   Eigenproteins, which correspond to the first principal component of a given module and serve as a summary expression profile for all proteins within a module, are shown for 10 of the 15 modules generated by WGCNA. Box plots depict mean (horizontal bars) and variance (25th to 75th percentiles), with whiskers extending to the last non-outlier measurements, are shown for all four groups (control, ALS, ALS/FTD, and FTD), and representative Gene Ontology (GO) terms are listed above each eigenprotein. Hub proteins for each of these modules are also highlighted. Significance was determined by comparisons of quantified proteins within individual cases across the clinical groups to the module eigenprotein using one-way nonparametric ANOVA Kruskal–Wallis with *P*-values listed above plots. Outlier cases are shown as open circles beyond the error bars.

B   Stacked bar plots represent analysis of differential expression (pairwise comparison between disease group and control) and enrichment of differentially expressed proteins within co-expression modules. Modules are listed along the *x*-axis, and the height of the bar along the *y*-axis indicates the proportion of differentially expressed module members, while the color indicates the fold change (red is increased, and blue is decreased) according to the scale shown.

▶

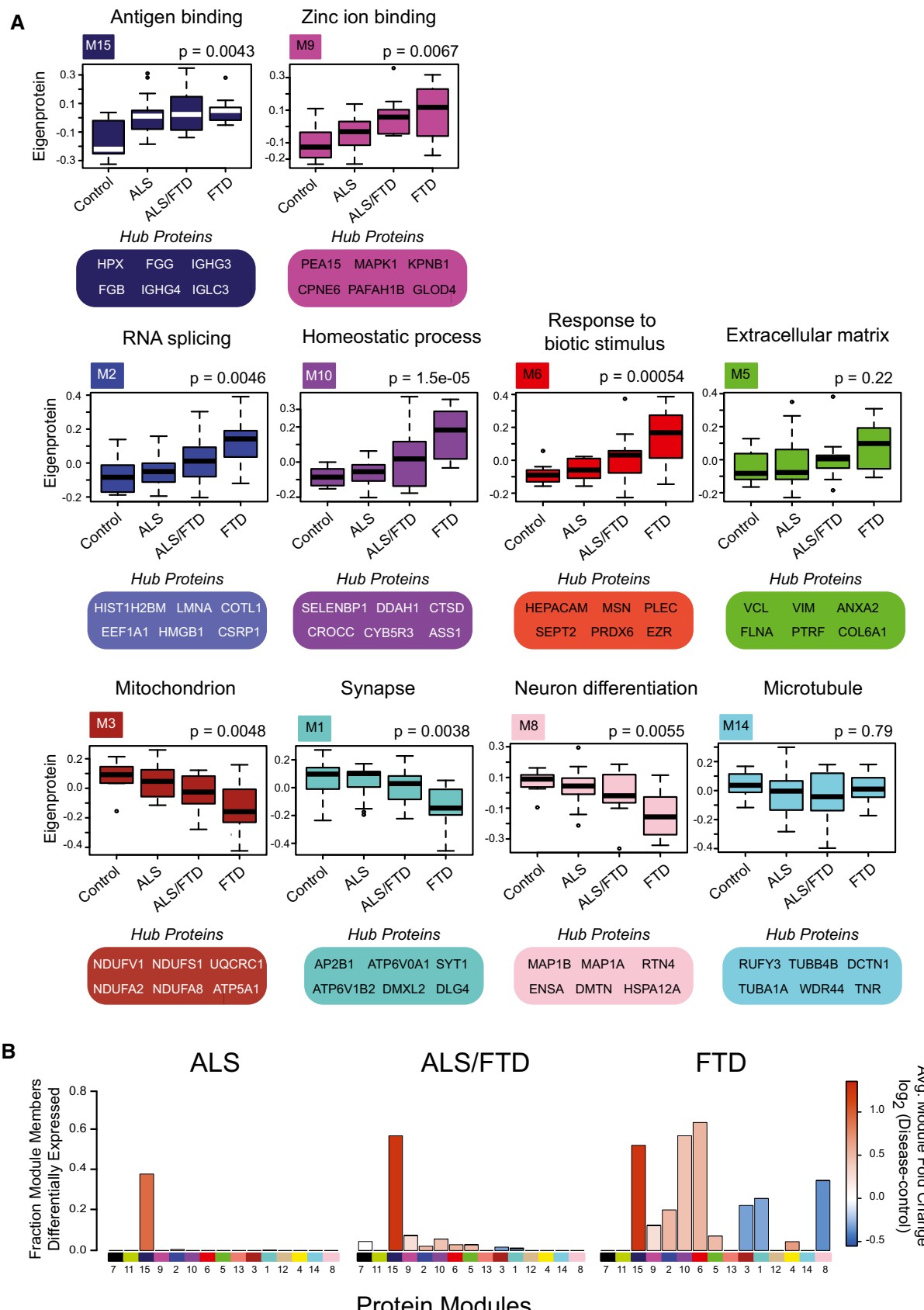

**Figure 2.**

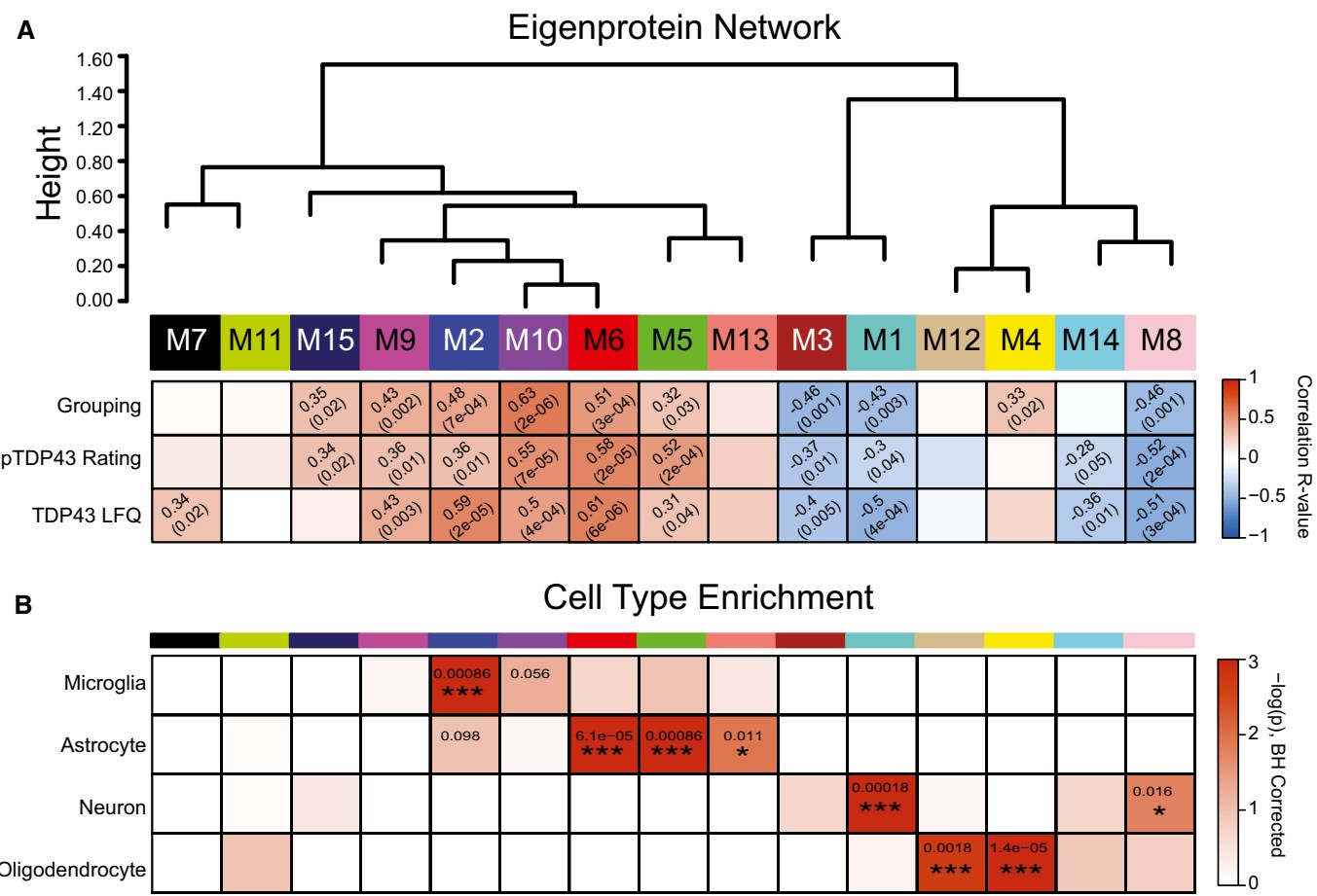

**Figure 3. Protein co-expression organizes the proteome into modules associated with brain cell types and clinicopathological traits.**

A WGCNA cluster dendrogram grouped proteins (*n* = 2,613) measured across the frontal cortex into distinct protein modules (M1–M15) defined by dendrogram branch cutting. Modules are clustered based on relatedness defined by correlation of protein co-expression eigenproteins (indicated by position in color bar). Listed in the heatmap are bicor correlations and *P*-values defining relationship between module eigenprotein expression and clinical grouping (defined as 0-control, 1-ALS, 2-ALS/FTD, and 3-FTD), pTDP-43 rating (defined as a score from 0 to 3 that represents pathological TDP-43 burden in the frontal cortex), and TDP-43 label-free quantification (LFQ).

B Cell-type enrichment analysis was performed using a one-tailed Fisher's exact test against lists of proteins previously generated from acutely isolated neurons, oligodendrocytes, astrocytes, and microglia. The heatmap displays Benjamin–Hochberg-corrected *P*-values (to control FDR for multiple comparisons) for the enrichment of certain cell types (vertical axis), and protein modules (horizontal axis) indicated by module number and color. Significance is demonstrated by the color scales, which go from 0 (white) to 3 (red), representing −log(*P*). Asterisks represent the level of significance of comparisons (**P* < 0.05; ****P* < 0.005).

and FTD (Fecto & Siddique, 2011), correlated with pathological TDP-43 scores and total TDP-43 levels. Additionally, M7, enriched with proteins involved in protein transport, correlated with TDP-43 pathology (Fig 3A).

As expected for the frontal cortex of ALS cases, where TDP-43 pathology is typically not found, and no cognitive decline phenotype exists, module eigenprotein expression for ALS patients without dementia was similar to controls. There was little overlap between controls and either FTD or ALS/FTD. Notably, we found differences in eigenprotein expression between ALS/FTD and FTD alone; namely, the eigenprotein expression for many of the modules representing the ALS/FTD group was intermediate between that of the strictly ALS and strictly FTD clinical groups, supporting the existence of a molecular spectrum of disease. Mapping of differentially expressed proteins to co-expression modules in cases with ALS/FTD and FTD further supports an

underlying molecular continuum (Fig 2B). There was an enrichment of differentially expressed proteins altered in pairwise comparisons between ALS/FTD and FTD patients and controls that were negatively correlated to neuronal modules (M1 and M8) and positively correlated to M2, M6, M9, and M10 modules. Additionally, the proteomic network generated in this study was compared to a previously generated proteomic network created from frontal cortex samples of control, ALS, AD, and PD cases from the Emory Brain Bank (Seyfried *et al*, 2017). Protein modules in both networks were highly preserved (Fig EV3A), and over-representation analysis revealed that all 15 modules generated in the current study had at least one cognate module within the previously generated Emory brain protein network (Fig EV3B). The consistency between these independent datasets provides confidence in the networks generated in this study, and an opportunity to expose meaningful relationships along the

ALS-FTD disease spectrum by relating clinical and pathological sample traits to modules.

## Protein co-expression classifies the proteome into modules associated with brain cell types

Co-expression organizes the proteome by placing proteins as nodes within a network with edges that define the connectivity, correlation, of nodes to one another (Gibbs *et al*, 2013). Many biological networks have an inherent hierarchical structure that allows nodes to be organized into a small number of highly interconnected modules (Parikshak *et al*, 2015). Intramodular connectivity creates a rank for proteins within a module, which allows identification of proteins that are module hubs; these are enriched for potential key drivers (Parikshak *et al*, 2015). Several modules were enriched for microglial and astrocytic proteins (M2, M10, M6, M5, and M13), while others were enriched for neuronal proteins (M1 and M8), or oligodendrocyte-specific proteins (M12 and M4) (Fig 3B), consistent with previous studies (Oldham *et al*, 2008; Miller *et al*, 2013; Seyfried *et al*, 2017). Modules of co-expressed proteins enrich with markers of specific brain cell types, and so changes in protein co-expression that relate to clinical phenotypes along the ALS-FTD disease spectrum may reflect changes in abundance, and/or activation of specific cell types (Seyfried *et al*, 2017). Module gene ontology enrichment also typically mirrors the cell-type enrichment analysis results (Dataset EV3). For example, the M6 module was enriched for proteins involved in responding to biotic stimuli consistent with the cell-type enrichment of astrocytic and microglial proteins with known roles in neuroinflammation. Modules enriched for gene ontologies associated with "synapse-part (GO:0044456)" and "neuronal differentiation (GO:0030182)", M1 and M8, respectively, are also enriched for neuronal-specific proteins. Microglial and astrocytic modules (M2, M6, M5, and M13) were positively correlated to pathological TDP-43 rating and clinical grouping, while neuronal modules (M1 and M8) were negatively correlated to these traits (Fig 3A). M4, enriched with oligodendrocyte proteins, also correlated moderately to ALS-FTD clinical grouping (Fig 3A). Together, this suggests cell-type-specific processes undergo changes in ALS/FTD that manifest as clinical and pathological endophenotypes, some of which are similar to changes occurring with AD (De Strooper & Karran, 2016; Seyfried *et al*, 2017).

## Module relevance to TDP-43 pathology and clinical grouping

The M2 and M6 modules, enriched for RNA binding proteins and inflammatory proteins, respectively, were strongly positively correlated to TDP-43 pathological burden, total TDP-43 levels and clinical grouping. TDP-43 mapped to M2, a module of co-expressed proteins increased significantly in ALS/FTD and FTD compared to controls. Notably, this module also included significant enrichment of microglial proteins (hypergeometric overlap *P* < 0.001), suggesting a strong co-expression between RNA binding protein dysfunction and microglial inflammation in the ALS-FTD spectrum that was not apparent for other neurodegenerative cohorts (Seyfried *et al*, 2017). As expected, immunohistochemical analysis of pTDP pathology showed a significant increase in FTD cases compared to ALS and control cases, which mirrors overall TDP-43 protein abundance in the frontal cortex of these brains (Fig EV2). Previous work has suggested that the

C-terminal peptide accumulation of TDP-43 is associated with potential toxicity of pathological TDP-43 (Zhang *et al*, 2009). Indeed, we observed an increase in C-terminal TDP-43 when analyzing peptide level differences (Fig EV2). Thus, the correlation of inflammatory proteins and RNA binding proteins in the M6 and M2 modules, respectively, to clinical and pathological traits, may represent a molecular pathophysiological connection between TDP-43 proteinopathy, related post-translational modifications, and consequent changes in the brain proteome occurring along the ALS-FTD disease spectrum.

## Modules associated with inflammation are enriched with TDP-43 interactors and have causal links to ALS

Modules associated with TDP-43 pathology and clinical phenotypes described above could either play a causal role in FTD or be secondary to the disease process. While postmortem protein expression does not directly assess causality of these modules by themselves, integrating multiple "omic" data sources can prioritize those modules that are most central to FTD and ALS pathogenesis. One such approach for module prioritization is to assess the enrichment of known TDP-43 protein–protein interactions (PPIs) within each respective module generated across the ALS-FTD spectrum of cases. For example, the M2 and M6 modules were significantly enriched for proteins previously identified (Freibaum *et al*, 2010) in studies exploring TDP-43 PPIs (Fig 4A). Furthermore, the protein products for several genes that have been causally linked to ALS (Taylor *et al*, 2016) are found within the M2 module, including HNRNPA1, MATR3, and PFN1 and TDP-43 itself (Fig 4B), whereas HSPB1 (in M6) has been linked to hereditary motor neuropathy, a form of motor neuron disease (James & Talbot, 2006; Rossor *et al*, 2011). Additionally, the M8 module, enriched with synaptic proteins, showed enrichment with TDP-43 PPIs, yet lacked any genes causally linked to ALS or FTD. The strong enrichment of TDP-43 interactors and other causal genes for ALS within M2 and M6 modules further reinforces their association with the ALS/FTD spectrum and suggests that proteins within these modules have critical roles in mechanisms that drive TDP-43 aggregation and other cellular-driven pathological processes (i.e., neuroinflammation).

## Astrocyte and microglial markers associated with neuroinflammation are increased across the ALS-FTD spectrum

The M6 module was significantly enriched for TDP-43 PPIs, and positively correlated to pathological TDP-43 burden and clinical dementia. This module includes hepatic and glial cell adhesion molecule (HEPACAM), membrane-organizing extension spike protein moesin (MSN), ezrin (EZR), glial fibrillary acidic protein (GFAP), and peroxiredoxin 6 (PRDX6). HEPACAM, an astrocyte-specific protein (Zhang *et al*, 2014), is involved in regulating cell motility and cell-matrix interactions, while moesin and ezrin are members of a family of proteins which function as cross linkers between plasma membranes and actin-based cytoskeleton, also regulating motility (Ivetic & Ridley, 2004). Many of the proteins in the M6 module are expressed in glial cell types (astrocytes and microglia), and overrepresent members of the gene ontology "response to biotic stimuli," thus defining what we characterize as the inflammatory module (Fig 5A). Module eigenprotein expression revealed M6 as a module increased in FTD compared to controls

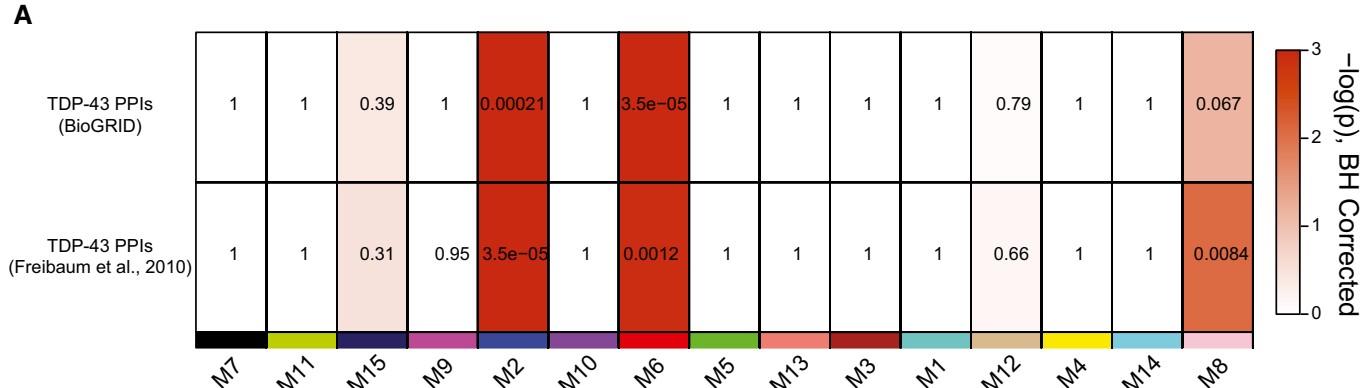

**A**

| | M7 | M11 | M15 | M9 | M2 | M10 | M6 | M5 | M13 | M3 | M1 | M12 | M4 | M14 | M8 |
|---|---|---|---|---|---|---|---|---|---|---|---|---|---|---|---|
| TDP-43 PPIs (BioGRID) | 1 | 1 | 0.39 | 1 | 0.00021 | 1 | 3.5e−05 | 1 | 1 | 1 | 1 | 0.79 | 1 | 1 | 0.067 |
| TDP-43 PPIs (Freibaum et al., 2010) | 1 | 1 | 0.31 | 0.95 | 3.5e−05 | 1 | 0.0012 | 1 | 1 | 1 | 1 | 0.66 | 1 | 1 | 0.0084 |

−log(p), BH Corrected

**B**

ALS GENES

TARDBP   MATR3

HNRNPA1 PFN1

HSPB1

**Figure 4. Modules with causal links to ALS are associated with inflammation and enriched with TDP-43 protein−protein interactions (PPIs).**

A    Enrichment of TDP-43 PPIs is displayed in this heatmap with results from TDP-43 PPI from BIOGRID and those from published global analysis of TDP-43 interacting proteins (Freibaum *et al*, 2010), with protein co-expression modules listed across the horizontal axis. Colors on heat map indicate enrichment (red) or no significant over-representation (white) for gene membership. Numbers displayed on the heatmaps represent positive signed −log$_{10}$(BH-adjusted *P*-values).

B    M2 (blue nodes) and M6 (red nodes) modules represented by proteins with high co-expression (gray lines) and TDP-43 PPIs (yellow lines). Several genes that have been previously linked to ALS-FTD are also highlighted by their colored module membership (inset).

and ALS. The WGCNA measure of intramodular connectivity (k$_{ME}$), defined as the Pearson correlation between the expression pattern of a protein and the module eigenprotein (which summarizes the characteristic expression pattern of proteins within a module), quantifies

the extent to which individual proteins mirror this pattern (Seidel *et al*, 2017). High relative k$_{ME}$ can be used to identify individual proteins that best represent a module; typically, these hubs of modules are markers of predominant cell types (Parikshak *et al*,

2015). Two astrocytic proteins (GFAP, HEPACAM) and two microglial proteins (MSN and TPP1) with high $k_{ME}$ were confirmed to be increased in abundance in frontal cortex homogenates from FTD cases compared to that of control and ALS cases by Western blot analysis (Figs 5B and EV4A and B). To assess cellular localization, a representative ALS and ALS/FTD case was immunostained with antibodies against GFAP, HEPACAM, and MSN and TPP1, which displayed astrocytic and microglial cellular populations, respectively, consistent with the specific cell-type expression of these protein markers (Fig EV4C). Although qualitative, the relative increase in tissue immunoreactivity seen for GFAP, HEPACAM, MSN, and TPP1 in the ALS/FTD case likely reflects changes in the abundance of astrocytes and microglia measured by mass spectrometry and confirmed by Western blotting. Thus, we found increased inflammation within the frontal cortex in FTD cases, but not ALS, which correlates with TDP-43 abundance, suggesting a potential role of glial cells in disease pathogenesis and progression related to TDP proteinopathy.

### Co-expression defines C9orf72 specific changes in ALS brain related to neuroinflammation

To elucidate the contribution of the C9orf72 genetic expansion to disease, we compared protein expression relative to our C9Pos case group. Few proteins were differentially expressed in the frontal cortex of C9Pos ALS cases compared to C9Neg ALS cases (Dataset EV2 and Fig 6A). Furthermore, none of these C9orf72 genotype-affected proteins were changed by greater than twofold, making it difficult to resolve proteome-wide differences by differential expression alone. The presence of the C9orf72 expansion as a case-sample trait also did not significantly correlate with any of the identified protein co-expression module eigengenes in the network when comparisons were drawn across all clinical groups. Together, these findings were unexpected, as we anticipated genetic contributions would generate a clear proteomic signature. However, mapping of the differentially expressed proteins in C9Pos ALS to the same modules from the co-expression network revealed that those proteins with significant, yet marginal, fold change in C9Pos ALS cases, generally mapped into the astrocytic/microglial modules (positively correlated with gene expansion) and neuronal modules (negatively correlated), consistent with changes associated with cognitive dysfunction in FTD (Fig 6B). Furthermore, an analysis, focused on strictly comparing only the non-demented ALS patients with and without the C9orf72 expansion, identified one module eigenprotein (M6) ($P < 0.05$) as increased in C9Pos ALS (Fig 6C). Notably, these patients were not demented prior to death, suggesting that the C9orf72 mutation may be associated with neuroinflammation in the brain. These results also highlight the ability of our integrative

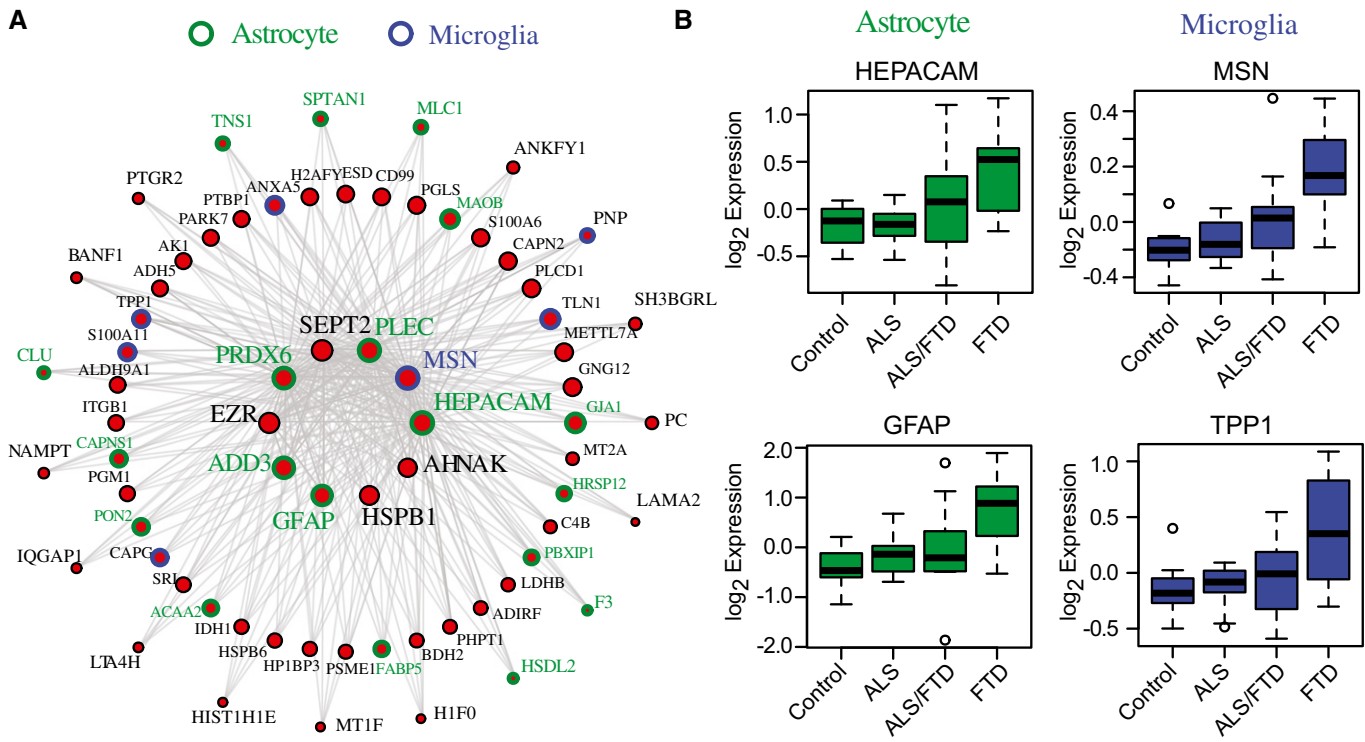

**Figure 5.  Astrocyte and microglial markers associated with neuroinflammation are increased across the ALS-FTD spectrum.**

A    I-graph of M6 module (red) representing hub proteins and corresponding gene symbols as nodes. Node size and edges (gray) are reflective of the degree of intramodular connectivity ($k_{ME}$). Specific cell-type expression is demonstrated by the node rim and text color around the gene symbols (blue: microglial marker and green: astrocytic marker).

B    Relative abundance measured by label-free quantification ($log_2$ expression values) for the astrocytic hub proteins HEPACAM and GFAP, and the microglial hub proteins MSN and TPP1 were increased in ALS/FTD and FTD cases consistent with the positive correlation of M6 to pathological TDP-43 pathological burden and clinical dementia. Box plots depict mean (horizontal bars) and variance (25th to 75th percentiles), with whiskers extending to the last non-outlier measurements, as shown.

systems approach to resolve genetic vulnerability, which was not confidently recognized through differential expression analysis alone. Although not significant ($P = 0.13$), we did observe a reduction in the 55-kDa-long isoform C9orf72 protein in C9Pos ALS cases compared to C9Neg ALS cases consistent with a loss of C9orf72 function (Fig EV5). This is of interest because haploinsufficiency of C9orf72 protein, possibly related to neuroinflammation, has been suggested as a possible disease mechanism in C9Pos patients (Farg *et al*, 2014; Waite *et al*, 2014; Yang *et al*, 2016).

## Discussion

Using an unbiased proteomic screen of the postmortem frontal cortex, we were able to identify and quantify protein differences along the ALS-FTD disease spectrum. Employing both differential

and co-expression analyses, we demonstrate that protein expression within the brain has a signature that defines the molecular and genetic underpinnings of the clinical and pathological ALS-FTD disease spectrum. WGCNA resolved related protein co-expression patterns or modules representing pathways and brain cell types, and showed a decrease in expression levels for modules associated with neurons and an increase in expression of astroglial and microglial modules associated with cognitive dysfunction and TDP-43 pathology in brain. Strikingly, C9orf72 expansion in patients dying with ALS predisposed the frontal cortex to elevated co-expressed markers of neuroinflammation, the same ones co-expressed in a concerted increased pattern across FTD and ALS/FTD patients. Thus, our findings in the context of genetic underpinnings of ALS/FTD and prior protein co-expression networks for AD (Seyfried *et al*, 2017) indicate that neuroinflammation is a common signature of dementia, while broad RNA binding protein co-accumulation is more

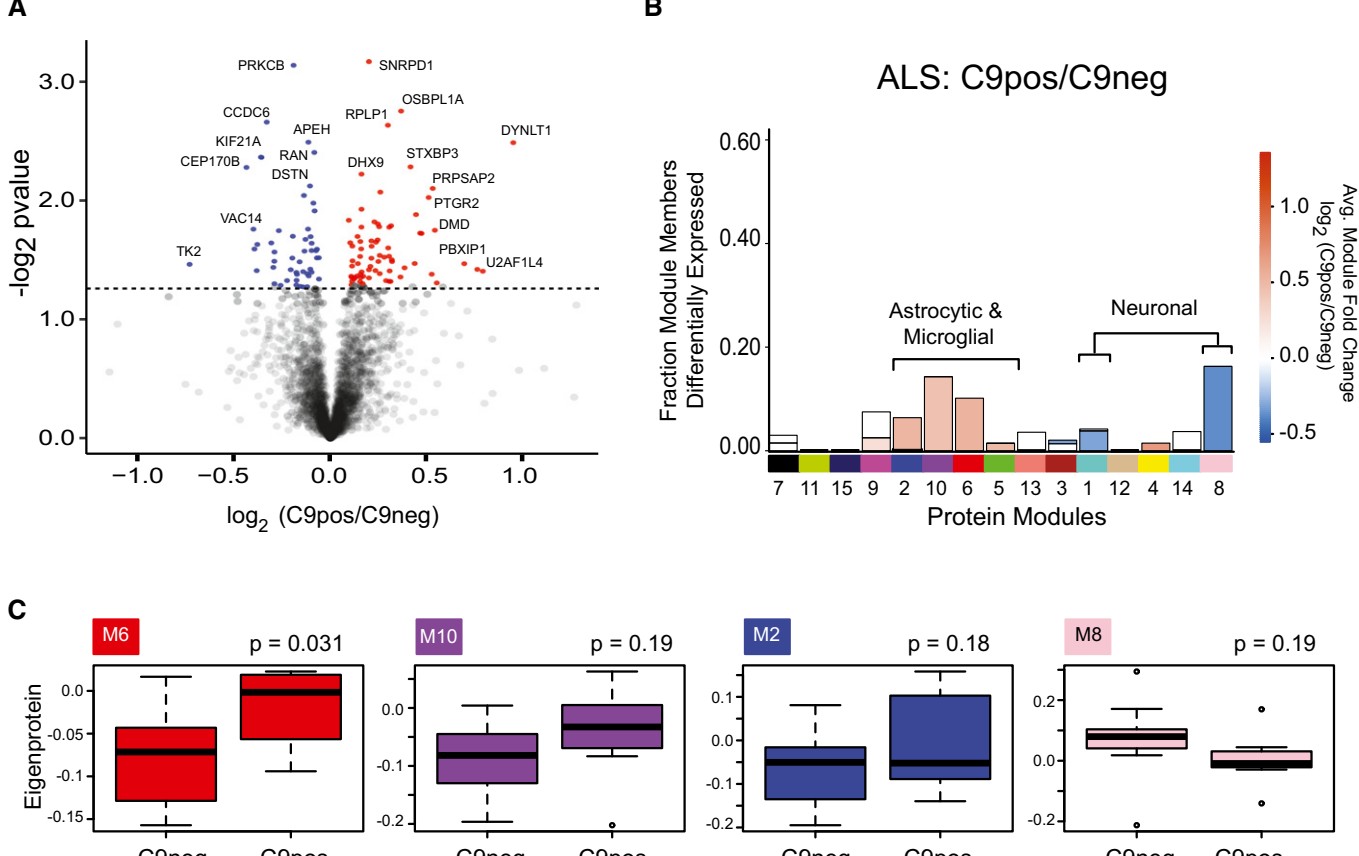

**Figure 6. Co-expression analysis resolves C9orf72-specific changes in ALS brain.**

A  Volcano plot displays the protein abundance (log$_2$ fold change) against the *t*-statistic ($-\log_{10}(P$-value)) for C9Pos ($n = 8$) against C9Neg ($n = 11$) ALS cases. Red and blue dots indicate significantly altered proteins that are increased and decreased, respectively, in C9orf72 carriers. Gray dots below the hashed line represent proteins that are not significantly changed ($P > 0.05$). Notably, there are no significant proteins that are twofold or more ($\pm$log$_2$ 1.0) increased or decreased in C9Pos cases compared to C9neg cases.

B  Stacked bar plot represents the analysis of differential expression (pairwise comparison between C9Pos and C9Neg ALS cases) and enrichment of differentially expressed proteins within co-expression modules. Modules are listed along the *x*-axis, and the height of the bar along the *y*-axis indicates the proportion of significantly differentially expressed module members, while the color indicates the fold change according to the scale shown.

C  Module eigenprotein (the summary expression profile of a module) for 4 of the 15 modules generated by comparing C9Pos ALS and C9Neg ALS cases. The M6 module is significant, whereas other glial modules (M10 and M2) and a synaptic module (M8) are increased and decreased in C9Pos cases, respectively, yet do not reach significance using a Student's *t*-test (*P*-value listed above the plot). Box plots depict mean (horizontal bars) and variance (25$^{th}$ to 75$^{th}$ percentiles), with whiskers extending to the last non-outlier measurements, as shown.

pronounced in the current network organized around frontal cortex of cases on the ALS-FTD clinical spectrum.

The protein co-expression network generated in this study was consistent with those previously reported in AD (Seyfried *et al*, 2017), which revealed biologically relevant modules linked to specific cell types and organelles (e.g., mitochondria). Although the genetic and pathological drivers in the ALS-FTD spectrum are distinct from AD, we observed a consistent downregulation of modules associated with neurons and synapses and upregulation of glial (microglial and astroglial) modules with increased TDP-43 pathology and cognitive dysfunction in brain. This suggests signatures reflecting relative changes in cellular phenotypes and/or abundance are shared across neurodegenerative diseases (De Strooper & Karran, 2016). However, one clear distinction between the AD network and our current ALS-FTD network was the definition of the M2 module that was enriched with both RNA binding proteins and microglial markers. This contrasts with AD networks in which the microglial and RNA binding proteins segregated to distinct modules (Seyfried *et al*, 2017). M2 showed a strong correlation with clinical and pathological traits of cases on the ALS/FTD spectrum, co-expressed with protein products of genes with causative links to ALS (hnRNPs, matrin 3, profilin 1, and TDP-43) and had significant enrichment of TDP-43 PPIs. The strong enrichment of TDP-43 inter-actors and other causal genes for ALS within this module further supports a strong association of M2 with the ALS/FTD spectrum and implicates other novel members of this module as having critical roles in TDP-43 biology that potentially influence TDP-43 aggregation or other pathological processes (i.e., microglial-directed neuroinflammation) inherent to ALS and FTD etiology.

Recent findings have demonstrated links between the C9orf72 expansion mutation and inflammatory pathways in C9orf72 animal models (Burberry *et al*, 2016; O'Rourke *et al*, 2016; Sudria-Lopez *et al*, 2016). Evidence also exists that C9orf72 mutation carriers have an increased prevalence of certain autoimmune disorders (O'Rourke *et al*, 2016), increased microglial pathology (Brettschneider *et al*, 2012), and increased thinning in frontal and temporal lobes in neuroimaging studies, compared to sporadic ALS and controls (Floeter *et al*, 2016). Increased abundance of proteins identified within the inflammatory co-expression module in the frontal cortex of ALS/FTD, FTD, and C9Pos ALS cases may explain the clinical link between cognitive decline and the C9orf72 expansion (Irwin *et al*, 2013; Umoh *et al*, 2016). Thus, from an unbiased proteomic approach, these data support the hypothesis that the dementia phenotype represents changes within the brain that correlate with, and are potentially causally linked to, increased neuroinflammation (Bettcher & Kramer, 2013).

There were several limitations in this work. First was the coverage of the proteome that we were able to achieve with our "single-shot" LC-MS/MS approach. The number of proteins identified was similar to proteomic coverage seen in other studies using similar approaches (Bi *et al*, 2017; Seyfried *et al*, 2017). However, this is much lower than studies investigating the transcriptome. Nevertheless, the co-expression patterns measured were robust, reproducible, and correlated with clinicopathological phenotypes. Another limitation was the relatively small number of FTD cases carrying the C9orf72 expansion; this diminished our power to discern any differences we may have seen due to this genetic mutation between ALS and FTD clinical groups. Future studies that

include additional C9orf72 expansion-positive cases from ALS/FTD and pure FTD cases will be critical at resolving genetic drivers of disease. Of interest, though, is our finding of increased microglia- and astrocyte-specific proteins in the frontal cortex of C9Pos ALS patients with no clinically detectable dementia.

Our results demonstrate the utility of a systems biology approach in understanding complex diseases with underlying comorbidity. We were able to relate a proteomic signature of disease (i.e., proteo-type) to clinicopathological phenotypes and C9orf72 genotype, which has not previously been done. Proteins within co-expression modules could further serve as potential biomarkers of disease mechanism or targets to assess therapeutic interventions across the ALS-FTD disease spectrum. Ultimately, this proteomic study provides a resource that moves towards a broad and comprehensive understanding of specific pathways, including inflammation, and cell-type-driven processes that relate to the clinical and genetic underpinnings along the ALS-FTD disease spectrum.

# Materials and Methods

### Case details

All brain tissues were obtained from the Emory Alzheimer's Disease Research Center (ADRC) Brain Bank. Human postmortem tissues were acquired under proper Institutional Review Board (IRB) protocols with informed consent from the subject or the family of the deceased subject, and all experiments conformed to the principles set by the WNA Declaration of Helsinki and the Department of Health and Human Services Belmont Report. Cases were selected based on clinical diagnoses. FTD cases were also selected based on neuropathological diagnoses to exclude FTD subtypes that did not have neuropathological diagnosis of FTLD-TDP (frontotemporal lobar degeneration characterized by ubiquitin and TDP-43-positive, tau-negative, FUS-negative inclusions) pathology. Emory neurologists cared for these patients throughout their disease course, and diagnoses were made using established clinical criteria (Brooks, 1994; Brooks *et al*, 2000; McKhann *et al*, 2001; Rascovsky *et al*, 2011). Standard diagnostic neuropathological analysis was performed for all cases, including phosphorylated TDP-43 (pTDP-43) immunohistochemistry on paraffin-embedded tissue sections using the phosphorylated Ser409/410 antibody (Cosmo Bio). Ordinal scales were used to assess pTDP-43 pathology (0–3) with higher scores indicating greater pathology. The presence of a C9orf72 repeat expansion was assessed from blood samples using the published repeat primed PCR method (DeJesus-Hernandez *et al*, 2011). Clinical and pathological information from all cases, including disease status, neuropathological criteria, age, sex, and postmortem interval are provided in Dataset EV1. Notably, 6 of 19 ALS cases and 1 of 10 control cases processed in this study overlapped with our previous study (Seyfried *et al*, 2017).

### Homogenization and proteolytic digestion of frontal cortex tissue

Dorsolateral prefrontal cortex (Brodmann area 9) tissue samples were processed essentially as previously described (Seyfried *et al*, 2017). In brief, each piece of tissue was individually weighed

(~100 mg) and homogenized in 500 µl of urea lysis buffer (8 M urea, 100 mM NaHPO$_4$, pH 8.5), including 5 µl (100× stock) HALT protease and phosphatase inhibitor cocktail (Thermo Fisher, Catalog #78440). Homogenization was performed using a Bullet Blender (Next Advance) following manufacturer's protocols. Each tissue piece was added to the urea lysis buffer in a 1.5-ml Rino tube (Next Advance) that contained 750-mg stainless steel beads (0.9–2 mm in diameter) and blended twice for 5-min intervals in a 4°C cold room. Protein supernatants were transferred into 1.5-ml Eppendorf tubes and sonicated (Sonic Dismembrator, Fisher Scientific) for three cycles of 5 s (at 30% amplitude) with 15-s intervals of rest to shear DNA. Samples were then centrifuged for 2 min at 15,871 g at 4°C. Protein concentration was assessed using the bicinchoninic acid (BCA) method, and samples were frozen at −80°C until use. Each homogenate was analyzed by SDS–PAGE to assess protein integrity. Brain protein homogenates (100 µg) were diluted with 50 mM NH$_4$HCO$_3$ to a final concentration of < 2 M urea. Samples were subsequently treated with 1 mM dithiothreitol (DTT) at 25°C for 30 min and then 5 mM iodoacetamide (Engelhart et al, 2004) at 25°C for 30 min in the dark. Protein samples were digested with 1:100 (w:w) lysyl endopeptidase (Wako) at 25°C for 2 h and then further digested with 1:50 (w/w) trypsin (Promega) overnight at 25°C. Resulting peptides were desalted with a Sep-Pak C18 column (Waters) and dried under vacuum.

## Liquid chromatography coupled to tandem mass spectrometry (LC-MS/MS)

For LC-MS/MS analysis, the peptides were first resuspended in 100 µl of loading buffer (0.1% formic acid, 0.03% trifluoroacetic acid, 1% acetonitrile). Peptide mixtures (2 µl) were separated on a self-packed C18 (1.9 µm Dr. Maisch, Germany) fused silica column (25 cm × 75 µM internal diameter (ID); New Objective, Woburn, MA, USA) by a Dionex Ultimate 3000 RSLCNano and monitored on a Fusion mass spectrometer (ThermoFisher Scientific, San Jose, CA, USA). Elution was performed over a 140-min total gradient at a rate of 300 nl/min with buffer B ranging from 1 to 65% (buffer A: 0.1% formic acid in water, buffer B: 0.1% formic in acetonitrile). The mass spectrometer cycle was programmed to collect at the top speed for 3-s cycles. The MS scans (400–1,600 m/z range, 200,000 AGC, 50 ms maximum ion time) were collected at a resolution of 120,000 at m/z 200 in profile mode and the HCD MS/MS spectra (0.7 m/z isolation width, 30% collision energy, 10,000 AGC target, 35 ms maximum ion time) were detected in the ion trap. Dynamic exclusion was set to exclude previous sequenced precursor ions for 20 s within a ±10 ppm window. Precursor ions with +1, and +8 or higher charge states were excluded from sequencing.

## Label-free protein quantification

Raw data files were analyzed using MaxQuant v1.5.2.8 with Thermo Foundation 2.0 for RAW file reading capability essentially as described with slight modifications (Seyfried et al, 2017). The search engine Andromeda was used to build and search a concatenated target-decoy Uniprot human reference database. Protein methionine oxidation (+15.9949 Da) and protein N-terminal acetylation (+42.0106 Da) were variable modifications (up to five allowed per peptide); cysteine was assigned a fixed carbamidomethyl

modification (+57.0215 Da). Only fully tryptic peptides were considered with up to two miscleavages in the database search. A precursor mass tolerance of ±10 ppm was applied prior to mass accuracy calibration and ±4.5 ppm after internal MaxQuant calibration. Other search settings included a maximum peptide mass of 6,000 Da, a minimum peptide length of six residues, 0.6 Da tolerance for ion trap HCD MS/MS scans. The false discovery rate (FDR) for peptide spectral matches, proteins, and site decoy fraction were all set to 1%. The label-free quantitation (LFQ) algorithm in MaxQuant (Luber et al, 2010; Cox et al, 2014) was used for protein quantitation as previously described. To account for possible confounds in run time, a brain peptide standard, generated from pooled samples of homogenized brain, was included at different points in the run set to control for drift over time and highlight consistency in the protein measurements (Fig EV1A).

## Data analysis and pre-processing

### Protein filtering and data imputation

Protein abundance was determined by peptide ion-intensity measurements across LC-MS runs using the label-free quantification (LFQ) algorithm in MaxQuant (Cox et al, 2014). In total, 47,977 peptides mapping to 4,178 protein groups were identified. However, one limitation of data-dependent label-free quantitative proteomics is missing quantitative measures, especially for low-abundance proteins (Karpievitch et al, 2012; Seyfried et al, 2017). Thus, only those proteins quantified in at least 90% of samples were included in the data analysis. After filtering, only allowing 10% missing values maximum across the 51 LC-MS/MS runs, 2,612 unique proteins were identified and robustly quantified (Fig EV1B). The 10% or fewer missing protein LFQ values were imputed using the k-nearest neighbor imputation function in R impute::impute.knn() function, similar to what has been previously described (Seyfried et al, 2017).

### Outlier removal and regression

Prior to data analysis, outlier removal was performed using Oldham's "SampleNetworks" v1.06 R script (Oldham et al, 2008) as previously published (Seyfried et al, 2017). Two control and two FTD cases were removed from the 51 cases initially included (Fig EV1B). Bootstrap regression of the remaining 47-case LFQ intensity matrix that explicitly modeled case status category while removing covariation with age at death, gender, and PMI was done following principal component analysis (PCA) of the expression data to confirm appropriate regression of selected traits, both in the "SampleNetworks" graphical output and via an in-house R script for PCA Spearman correlation to the amassed traits for 47 all non-outlier cases. PCA visualized that the top five principal components had Spearman correlation ρ < 0.3 with any of these three regressed covariates, and < 0.02 after regression.

### Differential expression analysis

Differentially enriched or depleted proteins ($P \leq 0.05$) were identified by ANOVA comparing the four clinical groups (control, ALS, ALS/FTD, and FTD). Multidimensional scaling as implemented in the WGCNA R package (Langfelder & Horvath, 2008) was used to visualize separation of cases using a subset of 165 proteins which had at least two ANOVA Tukey pairwise comparisons of high significance ($P < 0.01$) among the six possible comparisons

among ALS, FTD, FTD/ALS, and control groups. In a separate analysis, differentially expressed proteins in C9orf72 expansion-positive (C9Pos) ALS versus C9orf72 expansion negative (C9Neg) ALS (excluding cases with coexisting dementia) were identified by *t*-test. Differential expression is presented as volcano plots, which were generated with the ggplot2 package in Microsoft R Open v3.3.2. Significantly altered proteins along with corresponding *P*-value are listed in Dataset EV2.

### Co-expression network analysis

Following previously described procedures of WGCNA (Seyfried *et al*, 2017), a weighted protein co-expression network was generated using this pre-processed protein abundance matrix, using the WGCNA::blockwiseModules() function with the following settings: soft threshold power beta = 4.5, deepsplit = 4, minimum module size of 12, merge cut height of 0.07, signed network with partitioning about medoids (PAM) respecting the dendrogram and a reassignment threshold of $P < 0.05$. Specifically, we calculated pairwise biweight mid-correlations (bicor, a robust correlation metric) between each protein pair and transformed that matrix into a signed adjacency matrix (Langfelder & Horvath, 2012). The connection strength of components within this matrix was used to calculate a topological overlap matrix, which represents measurements of protein expression pattern similarity across the set of samples in the cohort constructed on the pairwise correlations for all proteins within the network (Yip & Horvath, 2007). Hierarchical protein correlation clustering analysis by this approach was conducted using 1-TOM, and initial module identifications were established using dynamic tree cutting as implemented in the WGCNA::blockwiseModules() function (Langfelder *et al*, 2008). Module eigenproteins were defined, which represent the most representative abundance value for a module and which explain covariance of all proteins within a module (Miller *et al*, 2013). Pearson correlations between each protein and each module eigenprotein were performed; this module membership measure is defined as $k_{ME}$. Figure EV1B illustrates workflow for analysis.

### Module preservation and over-representation analyses

Module preservation was tested using the "modulePreservation" WGCNA R package function, using exactly 500 permutations comparing the frontal cortex proteomic network generated from samples in this study against a previously generated Emory human brain protein network, a similar network built with ALS and other neurodegenerative disease cases from the Emory brain bank (Seyfried *et al*, 2017). This analysis was to ensure that the modules were representative of frontal cortex specific networks. Additionally, over-representation analysis (ORA) was conducted using Fisher exact tests (two-tailed) between module membership of the ALS-FTD proteomic network versus the previous Emory brain protein network (Seyfried *et al*, 2017). ORA analysis uses gene set enrichment analysis with a two-tailed Fisher exact test with 95% confidence intervals by employing the R function "fisher.test" (Seyfried *et al*, 2017). To reduce false positives, Benjamini–Hochberg FDR adjustment of *P*-values corrected for multiple comparisons.

### Enrichment analyses

To characterize differentially expressed proteins and co-expressed proteins based on gene ontology annotation, we used GO Elite v1.2.5 as previously published (Seyfried *et al*, 2017), with pruned output visualized using an in-house R script (Dataset EV3). Cell-type enrichment was also investigated as previously published (Seyfried *et al*, 2017). Enrichment of TDP-43 PPI across co-expression modules was investigated by intersecting module proteins with lists of genes known to interact with TDP-43, and assessing significance of overlap using a one-tailed Fisher exact hypergeometric overlap test. TDP-43 PPI lists from BioGRID (https://thebiogrid.org/117003/summary/homo-sapiens/tardbp.html), and a previously published global analysis of TDP-43 interacting proteins (Freibaum *et al*, 2010) was used. The total list of identified protein groups was used as the background and the PPIs lists (Dataset EV4) were filtered for presence in the total proteins list prior to cross-referencing. After assessing significance of TDP-43 PPIs, *P*-values were corrected for multiple comparisons by the Benjamini–Hochberg method.

## Western blotting

Frontal cortex tissue homogenates in Laemmli sample buffer were resolved by SDS–PAGE [NuPAGE Bis-Tris (Life Technologies)]. Gels were transferred onto nitrocellulose membranes (Invitrogen) using the iBlot 7-min dry transfer blotting system (Thermo Fisher Scientific). Blots were blocked with TBS starting block buffer (Thermo Fisher Scientific) for 30 min at room temperature and then probed with primary antibodies diluted in 10% blocking buffer in PBS overnight at 4°C. The next day, blots were rinsed and incubated with secondary antibodies conjugated to fluorophores, Alexa Fluor680 goat anti-mouse IgG(H+L) or Alexa Fluor680 goat anti- Rabbit IgG (H+L) (Life Technologies), for 1 h at room temperature. Images were captured using an Odyssey Infrared Imaging system (LiCor Biosciences). Protein densitometry for relative quantification was performed using ImageJ open source software. Higher molecular weight protein isoforms were considered in the quantification for both TPP1 and HEPACAM as each has been shown to be glycosylated (Golabek *et al*, 2003; Moh *et al*, 2005). Antibodies used include TDP-43 (Proteintech, 10782-2-AP; 1:1,000), tripeptidyl peptidase 1 (TPP1) (Sigma-Aldrich, WH0001200M1; 3 μg/ml), GFAP (Millipore, MAB360; 1:1,000), C9orf72 (Abcam, ab183892; 1:1,000), moesin (MSN) (Abcam, ab50007; 1:2,500), hepatic and glial cell adhesion molecule (HEPACAM) (Abcam, ab130769; 1:1,000), and glyceraldehyde 3-phosphate dehydrogenase (GAPDH) (EMD Millipore, AB2302; 1:1,000) as a loading control.

## Immunohistochemistry

Paraffin-embedded sections of frontal cortex (8 μm thickness) were deparaffinized by incubation at 60°C for 30 min and rehydrated by immersion in graded ethanol solutions. Antigen retrieval was performed by microwaving slides in 10 mM citrate buffer pH 6.0 for 5 min and then allowing slides to cool to room temperature (RT) for 30 min. Peroxidase quenching was performed by incubating slides in a 3% hydrogen peroxide solution in methanol for 5 min at 40°C. Slides were then rinsed in Tris-Brij buffer (1 M Tris-Cl pH 7.5, 100 mM NaCl, 5 mM $MgCl_2$, 0.125% Brij 35). For blocking, sections were incubated in normal goat serum or normal horse serum (Elite Vectastain ABC kit), depending on the primary antibody species, for 15 min at 40°C. Sections were then incubated with primary antibodies (diluted in 1% BSA in Tris-brij 7.5) overnight at 4°C. The

## The paper explained

### Problem
Amyotrophic lateral sclerosis (ALS) and frontotemporal dementia (FTD) are distinct neurodegenerative diseases that overlap in clinical presentation, neuropathology, and genetic underpinnings (e.g., C9orf72), yet the molecular basis for this overlap is not well understood.

### Results
We performed the first comparative quantitative mass spectrometry-based proteomic analysis of human postmortem brain tissues revealing molecular signatures distinguishing clinicopathological phenotypes across the ALS-FTD spectrum of disease. A systems-level co-expression network approach generated modules or groups of highly correlated proteins associated with RNA binding, synaptic transmission, inflammation, and specific cell types (neuronal, microglial, and astrocytic), which showed strong correlation with neuropathology and cognitive dysfunction. Modules were also examined for their overlap with TDP-43 interacting proteins, revealing one module enriched with RNA binding proteins and other causal ALS genes. Compared to sporadic ALS patients, C9orf72-positive ALS cases showed increased levels for a protein module associated with astrocytic and microglial markers, supporting a hypothesis linking the C9orf72 mutation to neuroinflammation.

### Impact
Our systems-level analysis of the brain proteome offers new insights into the pathways and cell types underlying clinical phenotypes across the ALS-FTD spectrum and provides evidence that the C9orf72 mutation is associated with neuroinflammation in brain.

following day sections were incubated in biotinylated secondary antibody at 5 μl/ml (Elite IgG Vectastain ABC kit) for 30 min at 37°C and then incubated with the avidin-biotin enzyme complex (Vector Laboratories) for 30 min. Stains were visualized by incubation of DAB Chromogen (Sigma-Aldrich) for 5 min at RT. Slides were then dehydrated in an ethanol series and mounted with coverslips. Slides were analyzed using an Olympus BX51 microscope and imaged with an Olympus DP70 camera. Antibodies used include phosphorylated TDP-43 (pTDP-43) (Cosmo Bio 409/410, TIP-PTD-P02; 1:1,000 for IHC), TPP1 (Sigma-Aldrich, WH0001300M1; 3 μg/ml), GFAP (Millipore, MAB360; 1:1,000), moesin (Abcam, ab50007; 2 μg/ml), and HEPACAM (Abcam, ab130769; 5 μg/ml). Pathological sample traits were represented by the abundance of phosphorylated TDP-43 cytoplasmic inclusions in the frontal cortex (defined by the pTDP score) or label-free quantification of TDP-43 (Fig EV2).

### Data availability

All raw data, MaxQuant output files, and analysis code used in this publication are available from Synapse (www.synapse.org) via accession syn10142580.

**Expanded View** for this article is available online.

### Acknowledgements
We are grateful to the patients and families that donated tissue samples to the Emory University brain bank for their contributions to this study. We acknowledge Dr. Daniel Geschwind and Vivek Swarup (Department of Genetics, UCLA) for helpful comments and discussion. We also thank the clinical teams whose analyses and care contributed to the dataset. Support was provided by the Accelerating Medicine Partnership AD grant U01AG046161-02, the NINDS Emory Neuroscience Core (P30NS055077), and the Emory Alzheimer's Disease Research Center (P50AG025688). N.T.S. is supported by 5R01AG053960-02 and in part by an Alzheimer's Association (ALZ), Alzheimer's Research UK (ARUK), The Michael J. Fox Foundation for Parkinson's Research (MJFF), and the Weston Brain Institute Biomarkers Across Neurodegenerative Diseases Grant (11060). M.E.U. was also funded by a pre-doctoral T32 NINDS training grant 3T32NS007480-16.

### Author contributions
MEU: Designed and performed experiments, analyzed the data, and wrote original and final draft of manuscript. EBD: Designed, implemented, and performed computational analyses, and participated in manuscript drafting and editing. JD: Performed experiments and analyzed the data. DMD: Designed, implemented, and performed experimental analyses. JJL: Participated in manuscript drafting and editing. AIL: Participated in manuscript drafting and editing. MG: Designed, implemented, and performed experimental analyses. JDG: Designed and implemented analyses, and participated in manuscript drafting and editing. NTS: Designed and implemented analyses, and participated in manuscript drafting and editing. All authors contributed to and have approved the final manuscript.

### Conflict of interest
The authors declare that they have no conflict of interest.

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
