## [Review Process File · EMBO Molecular Medicine]

A proteomic network approach across the ALS-FTD disease spectrum resolves clinical phenotypes and genetic vulnerability in human brain

Mfon E. Umoh, Eric B. Dammer, Jingtong Dai, Duc M. Duong, James J. Lah, Allan I. Levey, Marla Gearing, Jonathan D. Glass and Nicholas T. Seyfried

Corresponding authors: Nicholas T. Seyfried Jonathan D. Glass, Emory University School of Medicine

Review timeline:

Submission date:	29 June 2017
Editorial Decision:	12 July 2017
Revision received:	06 September 2017
Editorial Decision:	20 September 2017
Revision received:	13 October 2017
Accepted:	20 October 2017

Transaction Report:

Editor: Céline Carret

1st Editorial Decision

12 July 2017

Thank you for the submission of your manuscript to EMBO Molecular Medicine. We have now heard back from the two referees whom we asked to evaluate your manuscript, as their comments are consistent and overlapping, we felt that we didn't need a third one.

You will see from the comments below that the referees are enthusiastic about the study and have suggestions and recommendations to further improve conclusiveness and clarity as well as increase general interest, which is important for our journal. Of relevance, we would like to encourage you to provide independent validation of the data in a few patients at least as suggested by referee 1 to strengthen the conclusions.

I look forward to receiving your revised manuscript.

***** Reviewer's comments *****

Referee #2 (Remarks):

Umoh et al., performed the global proteomic network analysis across ALS-FTD diseased spectrum based on LS/MS analysis using postmortem frontal lobe brain tissue followed by

weighted correlation network analysis (WGCNA). First, multidimensional scaling (MDS) revealed that clinical phenotypes associate with innate proteomic signatures, as samples segregated into clusters representing clinical groups. Second, co-expression analysis revealed that modules associated with RNA binding proteins, synaptic transmission, and inflammation had strong correlation with TDP-43 pathology and cognitive dysfunction. Finally, the authors showed the relationship between the presence of C9 mutations in ALS and neuroinflammation. Proteomic investigation using many postmortem brain tissues is appreciable and the quality of data analysis is high. However, several issues need to be resolved before publication for EMM.

Major issues

1. It is desirable if proteomic modules would be validated in an independent cohort which has been done in their previous work on AD (Seyfried et al., 2016). Otherwise, some of expression results should be validated in other cases which have not been included in LS/MS screening experiments. If the authors can add cases with FTLT-tau or FTLT-FUS to validation, the manuscript may have more impact to recruit interests of readers.
2. There is a significant difference of age between control and ALS cases. The authors need to describe it and explain whether it affects protein expressions.
3. The clinical data of cases is not sufficient. For instances, race should be specified in Table 1. If possible, Mini-Mental State examination (MMSE) and/or Frontal Assessment Battery (FAB) in the cases used should be shown.
4. The authors need to explain how they chose 3 representative cases for each group in WB of Fig. 5B. It would be better to show WB with more cases or as a different cohort. Similarly, immunohistochemistry of Fig. 5C should be demonstrated with multiple cases.
5. It is quite confusing that there are apparent differences of GFAP and MSN between C9 negative and positive in WB of Fig. 6D and E, although there were few proteins differentially expressed (less than 2-fold) in the frontal cortex of C9Pos ALS cases compared to C9Neg ALS cases in LC/MS analysis. The data of Fig. 6D and E is unconvincing so that the authors can omit them from the manuscript if they are not able to confirm the results with more cases.
6. It would be significantly of intrigue if the authors compare the profiles of FTD with those of AD especially in regard to inflammation. The authors have reported that many inflammatory modules were distinct in symptomatic AD cases in the previous reports (Seyfried et al., 2016).

Minor issues

1. Six ALS and one control cases have been overlapped in the second cohort of their previous study (Seyfried et al., 2016). It is necessary to describe it in the manuscript.
2. In Fig 5B and C, the order of proteins, MSN, TPP1 (IBA1), HEPACAM, and GFAP should be the same as in the manuscript.
3. The immunoblots of IBA1 and HEPACAM should be added to Fig. 6D if the authors still leave the results in the manuscript. The WB images in Fig. 6D should be cropped properly and the signals in WB should be quantified.
4. The IBA1 staining should be added to Fig. 6E.

Referee #3 (Remarks):

Mfon Umoh and colleagues employed mass spectrometry to investigate the protein signatures of FTD, ALS, and ALS/FTD prefrontal cortex compared to controls. As one would expect, they found the most significant changes for FTD vs Ctrl, and fewer changes for ALS vs Ctrl, as the FTD patients had more severe neurodegeneration in the prefrontal cortex. ALS/FTD cases seemed to be midway between ALS and FTD, supporting the theory that these diseases are on a clinical spectrum. They primarily analyzed the data using WGCNA, which groups proteins into modules based on co-correlation of protein abundance across samples. Reassuringly, these modules were associated with typical sources of biological coexpression, such as gene ontology and brain cell type.

Much of the significant findings appear to be due to changes in cellular composition

changes in disease vs controls due to neuron death and gliosis, a problem inherent in whole-tissue analysis. However, grouping by modules allowed the authors to identify which modules were strongly associated with a specific cell type, and therefore likely correlated to changes in the relative abundance of that cell type or changes in the expression profile of that cell type. As one would expect modules enriched in neuronal proteins were decreased in FTD relative to control, and those enriched in glial proteins were increased. The authors also found a small increase inflammatory modules in C9ORF72 expansion cases vs sporadic ALS cases.

The authors also provide informative correlations of the modules with TDP-43 pathology, total TDP-43 levels, and cognitive dysfunction. Some of the modules also showed an enrichment of TDP-43 interacting proteins, including some ALS-associated proteins in particular modules. It is particularly intriguing that the modules enriched with both TDP43-interacting proteins and glial proteins were increased in FTD compared to Ctrl, yet the module enriched with TDP43-interacting proteins and neuronal proteins were decreased. This finding may provide mechanistic insight into differential roles of TDP-43 interactors.

This paper provides a very interesting and in depth analysis of different molecular modules involved in disease and is an excellent resource for other researchers. I highly recommend this paper for publication, and have a few minor suggestions:

- Figure 2 might be improved by adding prominent gene ontology information for each module.
- The authors do some comparisons of these data to similar data collected from other neurodegenerative disorders. They mention many of the modules are the same and the enrichment of TDP-43 interacting protein is unique to FTD and ALS. However, I think it would be interesting to expand on these finding more and show which particular protein levels are different in ALS vs other neurodegenerative disorders. This could provide valuable mechanistic insight into disease by focusing on molecular changes that are not just broadly present during neurodegeneration.
- Since many ALS patients develop cognitive dysfunction, especially later in disease, the ALS patient prefrontal cortex may be quite similar to early-stage FTD prefrontal cortex. This could allow investigators to differentiate early stage cellular disruptions that are perhaps causative of neuron death from later stage effects of cellular death. A more in depth analysis of protein changes seen all disease cases, but not in controls could be interesting.

1st Revision - authors' response

06 September 2017

Referee #2

Comment 1:

It is desirable if proteomic modules would be validated in an independent cohort which has been done in their previous work on AD (Seyfried et al., 2016). Otherwise, some of expression results should be validated in other cases which have not been included in LC/MS screening experiments. If the authors can add cases with FTLD-tau or FTLD-FUS to validation, the manuscript may have more impact to recruit interests of readers.

Author Response:

The suggestion by the reviewer of adding cases with FTLD-tau and FTLD-FUS is certainly of interest. Indeed, a project investigating the differences between FTLD TDP, FTLD-Tau, and FTLD-FUS is currently underway in our laboratory as part of the Accelerated Medicine Partnership-Alzheimer's Disease (www.nia.nih.gov/research/amp-ad). We have recently completed LC-MS/MS analysis of more than 500 individual brain samples representing patient cases of various neurodegenerative diseases that will be released as part of this consortium in the near future. The focus of the current paper, however, is ALS, and the FTLD seen in conjunction with ALS is exclusively TDP-43 related. FTLD-Tau and FUS are not associated with ALS. Though differences in the FTLD pathologies is of interest, it is

beyond the scope of this current paper. As shown in Supplemental Figure 3, we do validate that all proteomic modules are persevered across human post-mortem tissues. This is evidenced by a significant positive Fisher two-tailed overlap result for each of the ALS-FTD modules to at least one of the previously published (Seyfried, Dammer, et al 2017) Emory cohort modules in panel B of Supplemental Figure 3.

Comment 2:

There is a significant difference of age between control and ALS cases. The authors need to describe it and explain whether it affects protein expressions.

Author Response:

The proteomic data presented has been regressed on age, sex and post-mortem intervals (PMI) to control for influences of these co-variables on the protein expression (see Methods and Supplemental Figure 1). Notably, following regression, principal component analysis (PCA) demonstrated that age, PMI and gender had no influence on the protein expression data. This was highlighted in the “*Outlier removal and regression section*” under the “*Data analysis and pre-processing*” header in the methods. To make this more evident to the reader, we now mention that we regress for co-variables directly in the results section to better highlight this key point. It should be noted that similar regression schema have been used in large-scale genome network analysis ¹ and our recent proteome network analysis in Alzheimer’s Disease (AD) tissues ² and are highly recommended when using human clinical analysis to ensure that differential expression or co-expression findings are not confounded ³.

Comment 3:

The clinical data of cases is not sufficient. For instances, race should be specified in Table 1. If possible, Mini-Mental State examination (MMSE) and/or Frontal Assessment Battery (FAB) in the cases used should be shown.

Author Response:

For the patients diagnosed clinically with FTD only, the diagnosis was made by experienced, board certified cognitive neurologists (Drs. Allan Levey and James Lah, both co-authors on this manuscript) based on detailed clinical assessments with longitudinal follow-up using all available clinical, neuropsychological, imaging, and other biomarker data. All FTD patients also had FTLTDP43 on pathological examination. The clinical diagnosis of FTD in patients with ALS was again made by an experienced, board-certified neurologist (Jonathan Glass; co-corresponding author) based on the published clinical criteria ⁴.

Comment 4:

The authors need to explain how they chose 3 representative cases for each group in WB of Fig. 5B. It would be better to show WB with more cases or as a different cohort. Similarly, immunohistochemistry of Fig. 5C should be demonstrated with multiple cases.

Author Response:

We agree with the reviewer that the 3 representative cases in Western blots (WB) and immunohistochemistry (IHC) do not adequately represent the quantitative result we wish to show. To address this, we added box plots of label-free quantification (LFQ) protein expression values for GFAP, TPP1, HEPACAM, and Moesin (MSN). These data include *all* cases in each clinical group, and show the relative increase in these proteins along the ALS-FTD disease spectrum. Moreover, we increased the number of cases analyzed by Western blot (from 12 to 30 total cases), with quantification and statistical analysis by ANOVA with Dunnett’s post-hoc test (Supplemental Figure 4). As expected we see comparable results between the proteomic and WB findings (Figure 5 and supplemental

Figure 4). We provide this as a new Supplemental Figure 4 and have reported these findings in the results section on page 23. The presentation of IHC images was not meant to show quantitative changes, and we have corrected that misrepresentation. Quantification of protein expression by WB provides a relatively unbiased analysis within the prefrontal cortex, whereas IHC can show localization of proteins in various cell types and regions of tissues. We added additional IHC images of cases with ALS and ALS/FTD for qualitative comparison of protein patterning and cellular localization in both grey and white matter of frontal cortex and have added this as a new Supplemental Figure 5. Although qualitative, the relative increase in tissue immunoreactivity seen for GFAP, HEPACAM, MSN and TPP1 in the ALS/FTD case by IHC likely reflects changes in the abundance of astrocytes and microglia measured by mass spectrometry and confirmed by WB.

Comment 4:

It is quite confusing that there are apparent differences of GFAP and MSN between C9 negative and positive in WB of Fig. 6D and E, although there were few proteins differentially expressed (less than 2-fold) in the frontal cortex of C9Pos ALS cases compared to C9Neg ALS cases in LC/MS analysis. The data of Fig. 6D and E is unconvincing so that the authors can omit them from the manuscript if they are not able to confirm the results with more cases.

Author Response:

We agree and have removed the western blots and IHC (6D and 6E) from the main manuscript.

Comment 5:

It would be significantly of intrigue if the authors compare the profiles of FTD with those of AD especially in regard to inflammation. The authors have reported that many inflammatory modules were distinct in symptomatic AD cases in the previous reports (Seyfried et al., 2016).

Author Response:

We thank the author for this comment and more adequately address this topic on page 26 in the Discussion:

“The protein co-expression network generated in this study was consistent with those previously reported in AD², which revealed biologically relevant modules linked to specific cells types and organelles (e.g. mitochondria). Although the genetic and pathological drivers in the ALS-FTD spectrum are distinct from AD, we observed a consistent downregulation of modules associated with neurons and synapses and upregulation of glial (microglial and astroglial) modules with increased TDP-43 pathology and cognitive dysfunction in brain. This suggests signatures reflecting relative changes in cellular phenotypes and or abundance are shared across neurodegenerative diseases⁵. However, one clear distinction between the AD network and our current ALS-FTD network was the definition of the M2 module, that was enriched with both RNA-binding proteins and microglial markers. This contrasts with AD networks in which the microglial and RNA binding proteins segregated to distinct modules². M2 showed a strong correlation with clinical and pathological traits of cases on the ALS/FTD spectrum, co-expressed with protein products of genes with causative links to ALS (hnRNPs, matrin 3, profilin 1, and TDP-43), and had a significant enrichment for TDP-43 PPIs. The strong enrichment of TDP-43 interactors and other causal genes for ALS within this module further supports its strong association with the ALS/FTD spectrum and implicates other novel members of this module as having critical roles in TDP-43 biology that potentially influence TDP-43 aggregation or other pathological processes (i.e. microglial-directed neuroinflammation) inherent to ALS and FTD etiology.”

Minor issues of Reviewer 2:

Minor issue #1.

Six ALS and one control case have been overlapped in the second cohort of their previous study (Seyfried et al., 20[17]). It is necessary to describe it in the manuscript.

Author Response:

We now acknowledge this overlap and include the information in the Methods and in Supplemental Table 1.

Minor issue #2.

In Fig 5B and C, the order of proteins, MSN, TPP1 (IBA1), HEPACAM, and GFAP should be the same as in the manuscript.

Author Response:

The original WBs have been replaced by box plots of proteomic data and WBs with an additional number of cases ($n=30$ cases) are shown in new Supplemental Figure 4.

Minor issue #3.

The immunoblots of IBA1 and HEPACAM should be added to Fig. 6D if the authors still leave the results in the manuscript. The WB images in Fig. 6D should be cropped properly and the signals in WB should be quantified.

Author Response:

As suggested by the reviewer, Figures 6D and 6E have been removed from the main manuscript.

Referee #3:

Comment 1:

Figure 2 might be improved by adding prominent gene ontology information for each module.

Author Response:

We thank the reviewer for this suggestion and the top Gene Ontology (GO) terms have been added to an updated Figure 2.

Comment 2:

The authors do some comparisons of these data to similar data collected from other neurodegenerative disorders. They mention many of the modules are the same and the enrichment of TDP-43 interacting protein is unique to FTD and ALS. However, I think it would be interesting to expand on these finding more and show which particular protein levels are different in ALS vs other neurodegenerative disorders. This could provide valuable mechanistic insight into disease by focusing on molecular changes that are not just broadly present during neurodegeneration.

Author Response:

We find very few proteomic differences (by differential expression) that are specific to ALS in this analysis of the dorsal lateral pre-frontal cortex (DLPC), which is to be expected in patients who do not have dementia. Thus, there is no clear expressed phenotype that would be expected to manifest in the molecular network related to cellular changes. Had we analyzed the spinal cords or the motor cortex in the ALS cases, then we would expect to see many changes compared to controls (and to FTD). In total, there are 5 proteins that are unique to ALS in the differential expression compared to FTD and FTD/ALS. These include 2 isoforms of CamK2G, FASN, PRKAcB, and albumin, (Supplemental Table 2) and they do not represent any specific biological pathway.

Comment 3:

Since many ALS patients develop cognitive dysfunction, especially later in disease, the ALS patient prefrontal cortex may be quite similar to early-stage FTD prefrontal cortex. This could allow investigators to differentiate early stage cellular disruptions that are perhaps causative of neuron death from later stage effects of cellular death. A more in depth analysis of protein changes seen all disease cases, but not in controls could be interesting.

Author Response:

Though the suggestion that cognitive dysfunction occurs late in ALS disease seems reasonable, our clinical experience does not support that conclusion. Indeed, many (if not most) patients with ALS and FTD present with cognitive symptoms early in their ALS disease or even before motor symptoms are recognized, and patients without dementia live longer than those with dementia.⁶ In fact, the majority of people with ALS never develop overt cognitive dysfunction, and we have no reason to believe that the ALS cases included in our analysis would have developed dementia if they had lived longer. That said, one module (M15), enriched for blood microparticles and circulating immunoglobulin complexes, showed increased protein expression in all disease groups (ALS, ALS/FTD and FTD) compared to controls, consistent with a common mechanism of blood brain barrier (BBB) breakdown in neurodegenerative diseases (Figure 2 and Page 17 of the results). This could support the notion that BBB breakdown is one of the early signs of neurodegeneration.

References

1. Zhang, B.; Gaiteri, C.; Bodea, L.-G.; Wang, Z.; McElwee, J.; Podtelezchnikov, A. A.; Zhang, C.; Xie, T.; Tran, L.; Dobrin, R.; Fluder, E.; Clurman, B.; Melquist, S.; Narayanan, M.; Suver, C.; Shah, H.; Mahajan, M.; Gillis, T.; Mysore, J.; MacDonald, M. E.; Lamb, J. R.; Bennett, D. A.; Molony, C.; Stone, D. J.; Gudnason, V.; Myers, A. J.; Schadt, E. E.; Neumann, H.; Zhu, J.; Emilsson, V., Integrated systems approach identifies genetic nodes and networks in late-onset Alzheimer's disease. *Cell* 2013, *153* (3), 707-720.
2. Seyfried, N. T.; Dammer, E. B.; Swarup, V.; Nandakumar, D.; Duong, D. M.; Yin, L.; Deng, Q.; Nguyen, T.; Hales, C. M.; Wingo, T.; Glass, J.; Gearing, M.; Thambisetty, M.; Troncoso, J. C.; Geschwind, D. H.; Lah, J. J.; Levey, A. I., A Multi-network Approach Identifies Protein-Specific Co-expression in Asymptomatic and Symptomatic Alzheimer's Disease. *Cell systems* 2017, *4* (1), 60-72.e4.
3. Parikshak, N. N.; Gandal, M. J.; Geschwind, D. H., Systems biology and gene networks in neurodevelopmental and neurodegenerative disorders. *Nature reviews. Genetics* 2015, *16* (8), 441-58.
4. McKhann, G. M.; Albert, M. S.; Grossman, M.; Miller, B.; Dickson, D.; Trojanowski, J. Q., Clinical and pathological diagnosis of frontotemporal dementia: report of the Work Group on Frontotemporal Dementia and Pick's Disease. *Archives of neurology* 2001, *58* (11), 1803-9.
5. De Strooper, B.; Karran, E., The Cellular Phase of Alzheimer's Disease. *Cell* 2016, *164* (4), 603-15.
6. Umoh, M. E.; Fournier, C.; Li, Y.; Polak, M.; Shaw, L.; Landers, J. E.; Hu, W.; Gearing, M.; Glass, J. D., Comparative analysis of C9orf72 and sporadic disease in an ALS clinic population. *Neurology* 2016.

2nd Editorial Decision

20 September 2017

Thank you for the submission of your revised manuscript to EMBO Molecular Medicine. We have now received the enclosed report from the referee who was asked to re-assess it. As you will see the reviewer is now supportive and I am pleased to inform you that we will be able to accept your manuscript pending editorial amendments:

1) Please carefully address the comments of the referee and provide a letter INCLUDING the reviewer's reports and your detailed responses to their comments (as Word file).

***** Reviewer's comments *****

Referee #2 (Remarks for Author):

The manuscript has been improved by answering the requested questions. The reviewer still asks a couple of requests.

1. Insufficient validation

In Comment 1, I requested that the secondary cohort was necessary to validate the results as the same group has shown in their previous work (Seyfried et al., 2016). Unfortunately, Supplementary Fig.3 just validated the proteomic modules themselves, but did not do the differential changes. Did the authors justify the methodological differences between the previous and present studies?

2. Insufficient clinical information

The authors have been using "FTD" instead of "FTLD" in the manuscript, thus it is necessary to provide the precise clinical information since FTD is a clinical term. The neurologists should routinely assess Mini-Mental State examination (MMSE) and/or Frontal Assessment Battery (FAB) in the ALS/FTLD or FTD-suspected cases. Please include them as well as race background of cases. Besides, it was confusing that the authors replied that all FTD patients also had FTLD-TDP43 on pathological examinations; however, 3 cases of FTD had "0" score of pTDP43 in Table 1.

2nd Revision - authors' response

13 October 2017

Referee #2 Critique #1

In Comment 1, I requested that the secondary cohort was necessary to validate the results as the same group has shown in their previous work (Seyfried et al., 2016). Unfortunately, Supplementary Fig.3 just validated the proteomic modules themselves, but did not do the differential changes. Did the authors justify the methodological differences between the previous and present studies?

Author Response

We acknowledge that we did not further validate these proteomic modules or changes across the ALS-FTD spectrum observed in a second cohort of tissues as we did in the Cell Systems paper (Seyfried et al 2017). However, proteomics analysis from a second independent cohort of ALS and FTLD-TDP cases are forthcoming from the accelerating medicine partnership (AMP) Alzheimer's Disease (AD) consortium and we are eager to compare and integrate these proteomic findings with the current Emory cohort. Nevertheless, the analysis of over 50 human tissues across the ALS-FTD clinicopathological spectrum provides an impressive initial view of protein co-expression and strong module preservation when compared against our previously generated human brain networks (Seyfried et al 2017). The consistency between these separate networks provides further confidence in the module-trait relationships generated in this present study. Of note, all tissues samples were homogenized and processed identically to our previous study (Seyfried et al 2017). The only methodological difference was the mass

spectrometry pipeline, in which samples in the present study were analyzed on an Orbitrap Fusion mass spectrometer, whereas samples in the previous study were analyzed on an QE-plus Orbitrap mass spectrometer. The label free quantification (LFQ) and database search software (MaxQuant) was identical across studies as was the normalization/regression approaches used to control for confounding variables of age, gender and PMI. Since the networks were well preserved, we are confident that the choice in the mass spectrometer used was not a factor in the differential or co-expression analysis.

Referee #2 Critique #2

The authors have been using "FTD" instead of "FTLD" in the manuscript, thus it is necessary to provide the precise clinical information since FTD is a clinical term. The neurologists should routinely assess Mini-Mental State examination (MMSE) and/or Frontal Assessment Battery (FAB) in the ALS/FTLD or FTD-suspected cases. Please include them as well as race background of cases. Besides, it was confusing that the authors replied that all FTD patients also had FTLD-TDP43 on pathological examinations; however, 3 cases of FTD had "0" score of pTDP43 in Table 1.

Author Response

We thank the reviewer for bringing this to our attention and have looked at the autopsy reports for all of the "zero" cases. Each of these cases were done prior to the discovery of TDP-43 as a component of ubiquitin positive inclusions. Each was re-examined with pTDP-43 IHC and found to have extensive TDP pathology. All were clinically diagnosed with FTD. The TDP score of zero likely is due to the fact that the frontal-cortical section examined, which is an 8 μ m thick piece of tissue, did not show the presence of inclusions. Many other sections of the brain did show inclusions, supporting the diagnosis of FTLD-TDP. Notably, the homogenized tissue used for proteomics reflects a significantly larger sample pool of tissue.

In response to the concern for the lack of clinical data in the diagnosis of FTD, we note that many of our cases were collected in the era when routine screening for FTD was not done in the clinic. (We now screen all patients in the ALS clinic for FTD, and diagnosis of FTD in the cognitive clinic is done by detailed neuropsychological testing and clinical consensus). The clinical diagnosis of FTD was made according to the clinical criteria described by McKhann, et al in 2001 (Archives of Neurology). All examiners are board certified neurologists and specialists in their fields. In the patients with FTD without ALS, cases were chosen that had TDP-43 pathology and thus were FTLD-TDP. For patients with ALS and FTD, only the clinical criteria were used since the FTD associated with ALS is always FTLD-TDP. Nonetheless, all ALS/FTD patients had TDP-43 pathology in the brain.

Corresponding Author Name: Nicholas Seyfried and Jonathan Glass

Manuscript Number: EMM-2017-08202